# FEDERATED MIXTURE OF EXPERTS

## ABSTRACT

Federated learning (FL) has emerged as the predominant approach for collaborative training of neural network models across multiple users, without the need to gather the data at a central location. One of the important challenges in this setting is data heterogeneity; different users have different data characteristics. For this reason, training and using a single global model might be suboptimal when considering the performance of each of the individual user's data. In this work, we tackle this problem via Federated Mixture of Experts, `FedMix`, a framework that allows us to train an ensemble of specialized models. `FedMix` adaptively selects and trains a user-specific selection of the ensemble members. We show that users with similar data characteristics select the same members and therefore share statistical strength while mitigating the effect of non-i.i.d data. Empirically, we show through an extensive experimental evaluation that `FedMix` improves performance compared to using a single global model while requiring similar or less communication costs.

## 1 INTRODUCTION

An ever-increasing amount of devices are being connected to the internet, sensing their environment, and generating vast amounts of data. The term federated learning (FL) has been established to describe the scenario where we aim to learn from the data generated by this "federation" of devices (McMahan et al., 2016). Not only does the number of sensing devices increase, but also their processing power is increasing continuously to the point that it becomes viable to perform inference and training of machine learning models on device. In federated learning, the goal is to learn from these client devices' data without collecting the data centrally, which naturally allows for more private exchange of information.

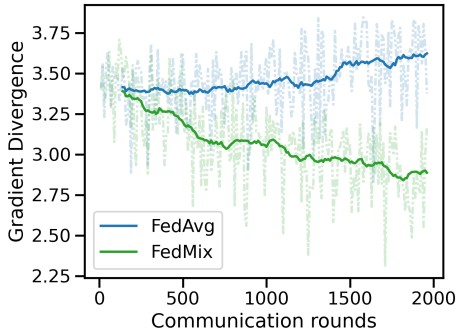

Figure 1: A sliding window of the gradient divergence (defined in Appendix D), on Cifar10 in the setup of Section 4 for `FedAvg` and `FedMix` ($K = 4$).

Several challenges arise in the federated scenario. Federated devices are generally resource-constrained, both in their computational capacity as well as in communication bandwidth and latency. In a practical example, a smartphone has limited heat dissipation capacity and must communicate via Wi-Fi. From a global perspective, devices' processing power and network connection can be highly heterogeneous across geographical regions and socio-economical status of device owners, causing practical issues (Bonawitz et al., 2019) and raising questions of fairness in FL (Li et al., 2019; Mohri et al., 2019).

One of the key challenges in FL that we aim to address in this work is the non-i.i.d nature of the shards of data that are distributed across devices. In non-federated machine learning, assuming independent and identically distributed data is generally justifiable and not detrimental to model performance. In FL however, each client performs a series of parameter updates on its own data shard to amortize the costs of communication. Over time, the direction of progress across shards with non-i.i.d data starts diverging (as shown in Figure 1), which can set back training progress, significantly slow down convergence and decrease model performance (Hsu et al., 2019).

To this end, we propose Federated Mixture of Experts (`FedMix`), an algorithm for FL that allows for training an ensemble of specialized models instead of a single global model. In `FedMix`, expert

models are learning to specialize in regions of the input space such that, for a given expert, each client's progress on that expert is aligned. `FedMix` allows each client to learn which experts are relevant for its shard and we show how it can be extended for inference on a previously unseen client. `FedMix` shows competitive performance against the established standard in FL, `FedAvg` (McMahan et al., 2016; Deng et al., 2020) across a range of visual classification tasks. Code will be released upon publication.

## 2 FEDERATED MIXTURE OF EXPERTS

Federated learning (McMahan et al., 2016) deals with the problem of learning a server model with parameters $\mathbf{w}$, *e.g.*, a neural network, from a dataset $\mathcal{D} = \{(\mathbf{x}_1, y_1), \dots, (\mathbf{x}_N, y_N)\}$ of $N$ datapoints that is distributed across $S$ shards, *i.e.*, $\mathcal{D} = \mathcal{D}_1 \cup \cdots \cup \mathcal{D}_S$, *without* accessing the shard specific datasets directly. By defining a loss function $\mathcal{L}_s(\mathcal{D}_s; \mathbf{w})$ per shard, the total risk can be written as

$$\arg\min_{\mathbf{w}} \sum_{s=1}^{S} \frac{N_s}{N} \mathcal{L}_s(\mathcal{D}_s; \mathbf{w}), \qquad \mathcal{L}_s(\mathcal{D}_s; \mathbf{w}) := \frac{1}{N_s} \sum_{i=1}^{N_s} L(\mathcal{D}_{si}; \mathbf{w}). \tag{1}$$

It is easy to see that this objective corresponds to empirical risk minimization over the joint dataset $\mathcal{D}$ with a loss $L(\cdot)$ for each datapoint. In federated learning one is interested in reducing the communication costs; for this reason McMahan et al. (2016) propose to do multiple gradient updates for $\mathbf{w}$ in the inner optimization objective for each shard $s$, thus obtaining "local" models with parameters $\mathbf{w}_s$. These multiple gradient updates are denoted as "local epochs", *i.e.*, amount of passes through the entire local dataset, with an abbreviation of $E$. Each of the shards then communicates the local model $\mathbf{w}_s$ to the server and the server updates the global model at "round" $t$ by averaging the parameters of the local models $\mathbf{w}^t = \sum_s \frac{N_s}{N} \mathbf{w}_s^t$. This constitutes federated averaging (`FedAvg`) (McMahan et al., 2016), the standard in federated learning.

One of the main challenges in federated learning is the fact that usually the data are non-i.i.d. distributed across the shards $S$, that is $p(\mathcal{D}|s_i) \neq p(\mathcal{D}|s_j)$ for $i \neq j$. On the one hand, this can make learning a single global model from all of the data with the classical `FedAvg` problematic. On the other hand, there is one extreme that does not suffer from this issue; learning $S$ individual models, *i.e.*, only optimizing $\mathbf{w}_s$ on $\mathcal{D}_s$. Although these individual models by definition do not suffer from non-i.i.d data, clearly we should aim to do better and exchange meaningful information between clients to learn more robust and expressive models.

### 2.1 THE FEDMIX ALGORITHM

With `FedMix`, we propose to strike a balance between the two aforementioned extremes; learning a single global model and learning $S$ individual models. For this reason, we revisit an old model formulation, the Mixture of Experts (MoE). The classical formulation of a MoE model (Jacobs et al., 1991; Jordan & Jacobs, 1994) contains a set of $K$ experts and a gating mechanism that is responsible for choosing an expert for a given data-point. A MoE model for a data point $(\mathbf{x}, y)$ can generally be described by

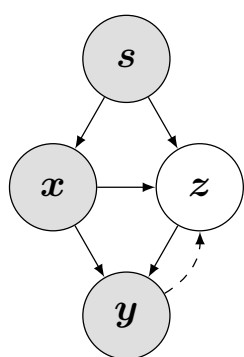

$$p_{\mathbf{w}_{1:K}, \theta}(y|\mathbf{x}) = \sum_{z=1}^{K} p_{\mathbf{w}_z}(y|\mathbf{x}, z) p_\theta(z|\mathbf{x}), \tag{2}$$

Figure 2: `FedMix` graphical model. The generative model is depicted with solid lines and the inference model with dashed lines.

where $z$ is a categorical variable that denotes the expert, $\mathbf{w}_k$ are the parameters of expert $k$ and $\theta$ are the parameters of the selection mechanism.

The MoE was proposed as a model for datasets where different subsets of the data exhibit different relationships between input $\mathbf{x}$ and output $y$. Instead of training a single global model to fit this relationship everywhere, each expert performs well on a different subset of the input space. The gating function models the decision boundary between input regions, assigning data-points from subsets of the input region to their respective experts.

In this work, we show that, in the federated scenario, sub-dividing the input region through a MoE can alleviate the consequences of non-i.i.d data by aligning gradient updates across experts (Figure 1). In Federated Mixture of Experts (`FedMix`) we enrich this model by conditioning the gating mechanism on the shard assignment $s$. Whatever characteristics make shard $s$ different from other shards can manifest in learning a different, localized gating mechanism that does not need to be communicated to the server. In choosing $K = 1$, `FedMix` recovers the standard setting of federated averaging. $K = S$ in combination with fixing $p(z = s|\mathbf{x}, s) = 1$ recovers $S$ independent models.

From a global perspective, we are interested in maximizing the following single objective:

$$\sum_{s=1}^{S}\sum_{i=1}^{N_s}\log p_{\mathbf{w}_{1:K},\theta_s}(y_{s,i}|\mathbf{x}_{s,i},s)=\sum_{s=1}^{S}\sum_{i=1}^{N_s}\log\Big[\sum_{z=1}^{K}p_{\theta_s}(z|\mathbf{x}_{s,i},s)p_{\mathbf{w}_z}(y_{s,i}|\mathbf{x}_{s,i},z)\Big] \quad (3)$$

Given the graphical model decomposition depicted in Figure 2, the objective in Eq. 3 corresponds to a federated MoE, where we have omitted the generative models $p(x|s)$. We will briefly touch upon the role of learning generative models in Appendix F but focus on the discriminative part of the model, *i.e.*, the MoE, in this paper.

While it is possible to optimize Eq. 3 directly, we have found empirically that it is hard to achieve both: avoiding collapse to a single expert, thus obtaining `FedAvg`, and specialization of the experts. Instead, we propose to form a variational lower-bound on Eq. 3 with a global variational approximation $q_\phi(z|\dots)$ to the true posterior $p(z|\mathbf{x}, y, s)$ with parameters $\phi$. At test time, $p(y|\mathbf{x}^*, s) = \sum_{k=1}^{K} p(y|\mathbf{x}^*, z)p(z|\mathbf{x}^*, s)$ can be readily evaluated without requiring $q$. This allows us to condition $q_\phi(z|\dots)$ on any available side-information *at training time* that might result in better specialization in the non-i.i.d federated scenario. In this paper we mainly consider classification tasks whose non-i.i.d nature predominantly stems from the non-i.i.d distribution of labels $y$. Other or additional known sources of misalignment could be included to further improve this approximation, such as a manufacturer-id for a medical device in a medical scenario, a geographic identifier, or general domain-specific information. We show one such additional example in 4.2. The lower bound to be maximized in `FedMix` therefore is as follows:

$$\sum_{s=1}^{S}\sum_{i=1}^{N_s}\log p_{\mathbf{w}_{1:K},\theta_s}(y_{s,i}|\mathbf{x}_{s,i},s) \quad (4)$$

$$\geq \sum_{s=1}^{S}\sum_{i=1}^{N_s}\sum_{z=1}^{K}q_\phi(z|y_{s,i})[\log p_{\mathbf{w}_z}(y_{s,i}|\mathbf{x}_{s,i},z)p_{\theta_s}(z|\mathbf{x}_{s,i},s) - \log q_\phi(z|y_{s,i})] \quad (5)$$

$$= \sum_{s=1}^{S}\sum_{i=1}^{N_s}\left(\sum_{z=1}^{K}q_\phi(z|y_{s,i})[\log p_{\mathbf{w}_z}(y_{s,i}|\mathbf{x}_{s,i},z)p_{\theta_s}(z|\mathbf{x}_{s,i},s)]\right) + H(q_\phi(z|y_{s,i})). \quad (6)$$

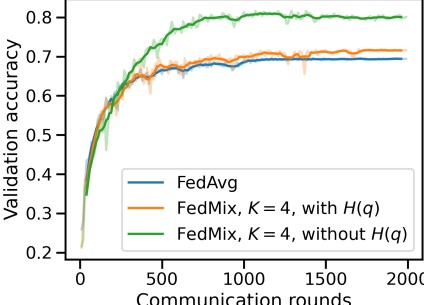

Figure 3: The effect of specialisation (without $H(q)$) compared to an ensemble (with $H(q)$) and `FedAvg` on Cifar10. The experimental setup is identical to what is described in Section 4.

Conditioning only on $y$ allows us to efficiently parameterize the variational approximation, incurring only a small communication overhead. While it would be possible to condition $q_\phi(z|y)$ on $s$, thus having localized approximations with parameters $\phi_s$ that do not need to be communicated, we found a global approximation to help align the gating mechanisms across shards. A global $q_\phi(z|y)$ encourages shards that contain data with the same label to assign them to the same expert.

Specialization of the experts is a key ingredient for `FedMix` to be successful; with specialization, the gradients for each expert become aligned across shards (see Figure 1), the hold-out accuracy improves (see Figure 3), and the communication costs decrease as each shard may only need to access a subset of the experts. We find that performing maximum a-posteriori (MAP) inference for $z$ generally leads to better and more personalized models. By removing the entropy term from equation 4, $q_\phi(z|y)$ and therefore $p_{\theta_s}(z|\mathbf{x}, s)$ are encouraged to concentrate and select only one expert for a

given data point. In the extreme case where a client's shard contains only data that is assigned to the same expert, we can reduce communication by receiving and sending updates for that single expert only. We show in Section 4 that communicating and evaluating experts based on thresholding the aggregate $q_\phi(z|s) = \mathbb{E}_{y \sim D_s}[q_\phi(z|y)]$ can reduce communication and computation overhead.

Figure 3 compares `FedMix` with and without the entropy term to standard `FedAvg` as a function of communication steps. With the entropy term, `FedMix` develops no expert specialization and collapses to an ensemble of $K = 4$ models. One drawback of the heavy specialization with MAP inference is that sometimes `FedMix` prematurely completely prunes experts, *i.e.*, $p_{\theta_s}(z = k|\mathbf{x}, s) \approx 0 \; \forall \mathbf{x}, s$. This can be undesirable as we lose model capacity that can be used for better modeling the data. As $q_\phi(z|y)$ is one of the main training signals of $p_{\theta_s}(z|\mathbf{x}, s)$, we introduce the marginal entropy term in the server, $H(E_{p(y)}[q_\phi(z|y)])$, as a regularizer. Notice that this leads to different training dynamics than locally optimizing the lower bound with the entropy included and we, empirically, found that it alleviates premature pruning, while still leading to specialized models. Figure 14a in Appendix H visualizes the development of $q_\phi(z|y)$ over time from initially uniform to high specialisation for the experiment depicted in Figure 3.

**Server Side Updates**   In a general federated learning algorithm, a central server selects a subset $S' \subset \{1, \dots, S\}$ of clients at time $t$ and transmits the current estimate of the global parameters $\mathbf{w}^t$ to them. These clients perform a series of mini-batch gradient updates with data from their shard $\mathcal{D}_s$ on a local loss function, which can come at the price of each client moving in possibly different directions in parameter space. In generalized `FedAvg` (Reddi et al., 2020), the server interprets $\Delta_s^t = \mathbf{w}^t - \mathbf{w}_s^{t+1}$ as a single-step gradient update from client $s$, averages those gradients and applies an optimizer such as Adam (Kingma & Ba, 2014) to receive $\mathbf{w}^{t+1}$. In light of non-i.i.d data across clients, this averaging strategy can result in slow progress since averaging updates in a highly non-convex parameter space can be sub-optimal. In `FedMix`, this effect is mitigated since for a given expert, the data that is used to update its parameters are aligned better across shards.

`FedMix` offers a second way to improve convergence speed by modifying the server-side updates. In generalized `FedAvg`, the individual gradients returned by the subset $S'$ of clients are averaged according to

$$\Delta^t = \sum_{s=1}^{S'} p(s) \cdot \Delta_s^t \, , \quad p(s) = \frac{N_s}{N_{S'}} . \tag{7}$$

In `FedMix`, we can speed up convergence by considering expert-specific updates $\Delta_{k,s}^t = \mathbf{w}_k^t - \mathbf{w}_{k,s}^{t+1}$. If a client $s$ pruned away expert $k$ from its local gating mechanism, $\Delta_{k,s}^t$ will be zero. We propose to normalize the effective magnitude of the resulting update $\Delta_k$ by up-weighing the updates of all other clients that do consider expert $k$ for their local mixture:

$$\Delta_k^t = \sum_{s=1}^{S'} p(s|z = k) \cdot \Delta_{k,s}^t \, , \quad p(s|z = k) \propto p(z = k|s)p(s) \, , \quad p(s) = \frac{N_s}{N_{S'}} . \tag{8}$$

Computing $p(z|s) = \mathbb{E}_{\mathbf{x} \sim \mathcal{D}_s}[p_{\theta_s}(z|s, \mathbf{x})]$ prior to sending updates to the server involves evaluating potentially large neural network models. Therefore we choose to approximate $p(z|s) \approx q_\phi(z|s) = \mathbb{E}_{y \sim \mathcal{D}_s}[q_\phi(z|y)]$, which involves just a single matrix multiplication. We discuss the implications of sending $q_\phi(z|s)$ to privacy and how these fare relative to `FedAvg` in Appendix C.

**Pruning experts**   Maintaining the cheap-to-compute marginal posterior per shard offers an additional opportunity to increase computation speed during local shard iterations and reduce overall communication costs. We propose to "prune away" experts locally from the MoE if $q_\phi(z|s)$ does not surpass a threshold $\eta/K$. In order to still optimize a valid bound, we need to re-normalize $q_\phi(z|y)$ before evaluation of the loss function (Pal et al., 2005). We evaluate the same threshold prior to sending updates to the server in order to avoid communicating parameters that have not changed during the client's iterations. Once the server selects a client for another round, it provides only those experts to the client that were updated by the client in the previous round. We empirically find that the entropy of $q(z|s)$ decreases steadily and we prune away experts $k$ with probability $q(z = k|s) < \eta/K$ without significant drop in performance. We explore the consequences of pruning experts in the experiment section and in Appendix E. Algorithm 2 shows how `FedMix` can be enriched by pruning.

**Algorithm 1** The `FedMix` algorithm. $\alpha, \beta$ are the client and server learning rates respectively

---

**function** SERVER SIDE
 Initialize $\phi$ and $K$ vectors $\mathbf{W} = [\mathbf{w}_1, \ldots, \mathbf{w}_K]$
 **for** round $t$ in $1, \ldots T$ **do**
  $S' \leftarrow$ random subset of the clients
  Initialize $\Delta_{\mathbf{W}}^t = \mathbf{0}, \Delta_{\phi}^t = \mathbf{0}$
  **for** $s$ in $S'$ **do**
   $\mathbf{W}_s^t, \phi_s^t, p(z|s) \leftarrow$ CLIENT SIDE$(s, \phi, \mathbf{W})$
  **end for**
  $p(s|z) \leftarrow p(z|s)p(s) / \sum_{s \in S'} p(z|s)p(s)$
  **for** $s$ in $S'$ **do**
   $\Delta_{\mathbf{w}_k}^t += p(s|z=k)(\mathbf{w}_k^{t-1} - \mathbf{w}_{s,k}^t) \ \forall k$
   $\Delta_{\phi}^t += \frac{N_s}{N_{S'}}(\phi^{t-1} - \phi_s^t)$
  **end for**
  $\Delta_{\phi}^t -= \nabla_{\phi} H(\sum_c q_{\phi}(z|y=c)p(y=c))$
  $\mathbf{w}_{1:K}^{t+1} \leftarrow$ ADAM$(\Delta_{\mathbf{w}_{1:K}}^t, \beta)$
  $\phi^{t+1} \leftarrow$ ADAM$(\Delta_{\phi}^t, \beta)$
 **end for**
**end function**

**function** CLIENT SIDE$(s, \phi, \mathbf{W})$
 Get local parameters $\theta_s$
 **for** epoch $e$ in $1, \ldots, E$ **do**
  **for** batch $b \in B$ **do**
   $L_s \leftarrow \mathbb{E}_{q_{\phi}(z|y_b)}[\log p_{\mathbf{w}_z}(y_b|\mathbf{x}_b, z)p_{\theta_s}(z|\mathbf{x}_b, s)]$
   $\phi += \alpha \nabla_{\phi} L_s$
   $\mathbf{W} += \alpha \nabla_{\mathbf{W}} L_s$
   $\theta_s += \alpha \nabla_{\theta_s} L_s$
  **end for**
 **end for**
 $q(z|s) \leftarrow \mathbb{E}_{y \sim \mathcal{D}_s}[q_{\phi}(z|y)]$
 **return** $\mathbf{w}_{1:K}, \phi, q(z|s)$
**end function**

**Designing robust gates** In the federated scenario, $N_s$ is often much smaller than $N$ and especially small in relation to the complexity of the data we try to model. Any localized parameters therefore are prone to overfitting. On the other hand, the global parameters of an expert are trained using all data-points assigned to that expert across all shards, allowing to learn more robust features.

We can make use of the robustness of these experts' features for the gating mechanism by conditioning on them instead of training an entirely separate model for $p_{\theta_s}(z|\mathbf{x}, s)$. Let us define $h_k(\mathbf{x})$ as intermediary features of expert $k$. Since not all experts might be used for a given shard and in order to scale with $K$, we average over the marginal posterior of the training set at that shard before applying a linear transformation to compute the input to the softmax gates:

$$h_s(\mathbf{x}) = \sum_{k=1}^{K} q_{\phi}(z=k|s)h_k(\mathbf{x})$$

$$p_{\theta_s}(z|\mathbf{x}, s) = \text{SM}\left(\mathbf{A}_s^T h_s(\mathbf{x}) + \mathbf{b}_s\right) \tag{9}$$

where $\theta_s = (\mathbf{A}_s, \mathbf{b}_s)$ are local learnable parameters and SM represents the softmax function.

**Inference at test time** We consider three variants for test-time evaluation of `FedMix`. In the first case, a client $s$ that participated in training is presented with a new data point $(\mathbf{x}^*, s)$. Predictions can then be straightforwardly done by selecting the $y$ that maximizes $\sum_{z=1}^{K} p(y|\mathbf{x}^*, z)p(z|\mathbf{x}^*, s)$. In the second, more challenging, scenario a new client $s^*$ is introduced together with a new labelled local data set $\mathcal{D}_{s^*}$. Here we propose to instantiate and train the local gating mechanism by optimizing the parameters $\theta_s$ of $p_{\theta_s}(z|\mathbf{x}, s^*)$ via MAP inference at the local objective. Afterwards, predictions can be made in a manner similar to the first case.

Finally, we consider the case in which a new client $s^*$ has no labelled dataset available. Without a local gating function, simply ensembling experts exhibits almost random behaviour since experts can be overly confident on out-of-distribution data (Snoek et al., 2019). We therefore propose to ensemble across local gating mechanisms to compute $p(z|\mathbf{x}^*) = \sum_{s=1}^{S} p_{\theta_s}(z|\mathbf{x}^*, s)p(s|\mathbf{x}^*)$; a method which works well in practice. In Appendix F we discuss results for new shard inference as well as a more principled approach which makes use of the graphical model formulation in Figure 2.

## 3 RELATED WORK

`FedMix` has similarities to many recent works in the topic of federated learning. Two methods closely related to ours are described in (Sattler et al., 2019; Briggs et al., 2020). The authors propose to perform hierarchical clustering on the updates returned from each shard in order to incrementally create separate models for groups of users, with a cluster assignment mechanism based on handcrafted heuristics. `FedMix` instead takes a different approach; it starts with a fixed set of $K$ models

and then optimizes with gradient descent at each shard a per-datapoint model assignment mechanism that can better fit the peculiarities of the local dataset.

Another closely related work is presented by Mansour et al. (2020), where the authors propose to similarly create an ensemble of $K$-models and assign to each shard the model that achieves the lowest training loss on the local dataset. This is closer to the assignment that happens in `FedMix` with one main difference; `FedMix` takes into account the uncertainty in the selection mechanism as well with $p(z|\mathbf{x}, s)$ instead of selecting the top performing component during training. This is beneficial early in training where the models have not fully specialized yet. Using local and global model parts has also been explored by Liang et al. (2020). The authors propose to have a local feature extractor at each shard and a global classifier on top of those features as opposed to having $K$-separate models and a local selection mechanism as in `FedMix`. This setup yields improvements upon the vanilla federated averaging algorithm, however there are two potential drawbacks; first, empirically, the authors had to start their procedure from a pre-trained model with `FedAvg` and secondly, they have to ensemble all of the different feature extractors for predictions in new shards. In our experiments, we omit the pre-training step and show that the ensembling strategy fails.

Federated learning in the non-i.i.d setting can also be improved upon in other ways. Li et al. (2018) propose to employ a proximal regularizer at the shard level in order to prevent the local models from drifting too far from the global model, thus making federated learning more robust. Jiang et al. (2019) notice that `FedAvg` and Reptile (Nichol et al., 2018), a meta-learning algorithm, are essentially the same algorithm and thus propose fine tuning with Reptile in order to improve the personalized performance of the global model. In a similar vein, there are promising new works that explore the meta-learning view of federated learning (Chen et al., 2018; Khodak et al., 2019; Fallah et al., 2020). Improving the personalized performance of the global model has also been done without meta-learning in works such as by Deng et al. (2020); Mansour et al. (2020). In general, such improvements are complementary to `FedMix` and can be used to further enhance its performance. We refer the interested readers to the recent surveys by (Kairouz et al., 2019; Kulkarni et al., 2020).

## 4 EXPERIMENTS

We evaluate `FedMix` on three datasets: Cifar10, Cifar100 (Krizhevsky et al., 2009) and Femnist (Caldas et al., 2018), a 62-way image classification problem on hand-written digits and letters that is naturally non-i.i.d due to different writing styles of 3500 users. In Appendix A we detail the experimental setup and provide additional ablation studies in Appendix E.

### 4.1 PERFORMANCE VS. COMMUNICATION

We compare `FedMix` along several dimensions to baselines such as (generalized) `FedAvg` (Reddi et al., 2020), *biased* `FedAvg`, and the Local/Global approach of Liang et al. (2020). In *biased* `FedAvg`, we allow each client to learn a personalized bias vector $b_s$ of its output layer. This will allow *biased* `FedAvg` to model the label skew at each client but, fundamentally, cannot model any other form of non-i.i.d-ness. Similarly, Liang et al. (2020) propose to split the model into local and global components by having local feature extractors and learning the upper layers of the neural network via `FedAvg`. We experimented with splitting LeNet-5 at every intermediate layer and report results with the best performing split: keeping the input layer local. For ResNet-20, splitting after the first block performed best. First, we show that training with `FedMix` achieves higher global model accuracy. Second, we show that for a given communication budget, `FedMix` can be competitive against baselines. Table 1 contains test set results for Cifar10, evaluated after $2k$ communication steps and Femnist averaged over the last $2k$ steps. Figures 4d, 4f reveal that the models have converged at that point. For Cifar100, we observe no overfitting and report test-set accuracies averaged over the last 10 evaluations. Figure 4 shows learning curves for different settings of `FedMix`. All of these results were obtained by using the server version of the global parameters; the results with additional local fine-tuning at the client level are discussed in Appendix G.

For **Cifar10** we can see that except for the initial training phase, `FedMix` improves over `FedAvg` for a given communication budget. For the same communication costs per communication round, `FedMix` offers better performance than `FedAvg`, as can be seen in comparing `FedMix` ($K = 2$) to

`FedAvg` LeNet-5 with 61 channels and `FedMix` ($K = 4$) to `FedAvg` LeNet-5 with 113 channels. The channel count has been chosen such that the model size approximates 2 and 4 separate standard LeNet experts respectively. Against *biased* `FedAvg` and Local/Global Liang et al. (2020), `FedMix` can improve significantly in terms of inference on a new shard (Figure 12 in Appendix F). When endowing `FedMix` with the ability to learn a single personalized bias vector $b_s$ across all experts, we can harness some of the advantages of *biased* `FedAvg` for inference on a new shard while retaining `FedMix`' advantage when performing inference on a new shard.

For **Cifar100**, we can see that learning progresses much faster for `FedMix` in terms of communication rounds than `FedAvg`. Against `FedAvg`, we can see that `FedMix` improves with both K=4 and K=10 on a per round progression, and with K=10 and expert pruning, it eventually provides a better accuracy and communication trade-off than `FedAvg`. *Biased* `FedAvg` seems to be the better inductive bias in this task, although one should be weary that its usefulness is limited to label skew and does not generalize to other sources of non-i.i.d.ness (see section 4.2). This is in contrast to `FedMix` which is designed to handle any type of explicit non-i.i.d.ness.

For **Femnist** we compare against `FedAvg` and observe that generally `FedMix` is more prone to overfitting due to the extra amount of model parameters and expert specialization. For this reason, we learn the parameters of the local gating networks via `FedAvg` across all clients. The gating network is still localized through the use of $q(z|s)$ in pooling of $h_k(\mathbf{x})$ (equation 9). We see that increasing $K$ improves the performance of `FedMix`. Since `FedMix` effectively requires the model to first identify the right expert, followed by evaluating that expert's prediction, compounding errors can lead to `FedMix` being outperformed by `FedAvg`, such as for $K = 2$. *Biased* `FedAvg` performs similar to standard `FedAvg` since the non-i.i.d-ness does not predominantly stem from label-skew.

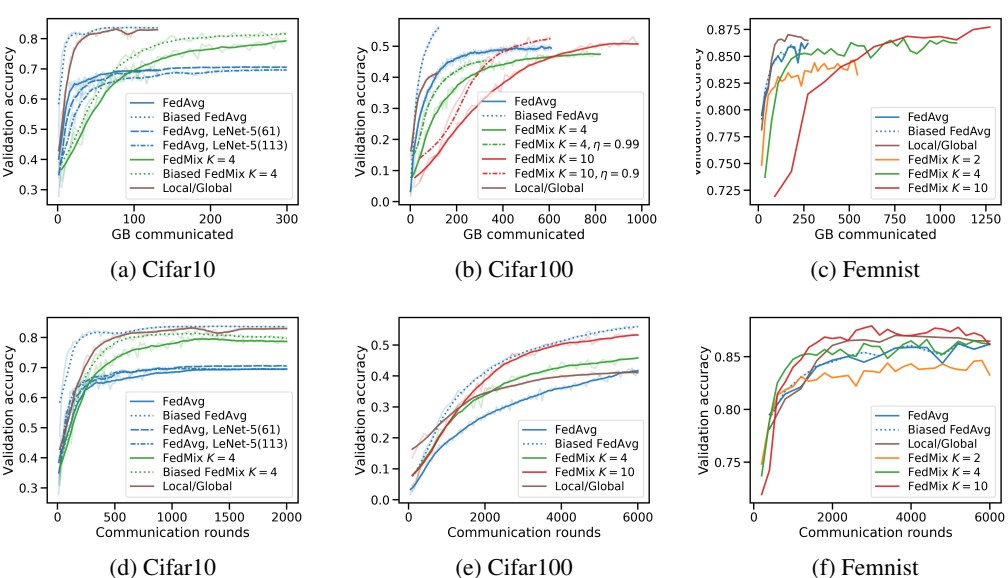

Figure 4: Average accuracy across all clients (y-axis) as a function of the amount of GB communicated (top row) and as a function of communication rounds (bottom row). Cifar 10 models are trained on the standard 45k training split. x-axes have been truncated for improved visibility. Best viewed in color.

## 4.2 ROTATED MNIST

To show that `FedMix` is not limited to label-skew, we create a federated rotated MNIST dataset with 100 clients. Instead of label skew, each client randomly chooses a multiple of 45 degrees

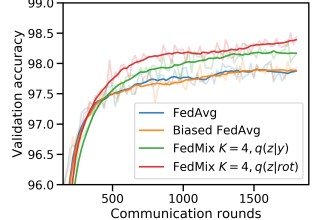

Figure 5: Only rotation

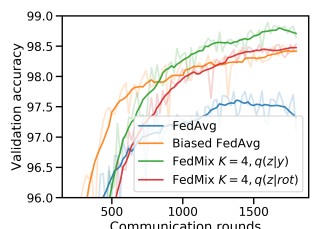

Figure 6: Rotation and labels

Table 1: Average test-set accuracies across clients and communication costs (rounds and GB) for Cifar10, Cifar100 and Femnist. For Cifar10 we report the performance after $2k$ rounds, for Cifar100 we report the average over the last 10 evaluations at the end of training and for Femnist we report the average over the last $2k$ communication rounds.

| Method | Cifar 10 | | Cifar 100 | | Femnist | |
|---|---|---|---|---|---|---|
| | Acc. | Comm. | Acc. | Comm. | Acc. | Comm. |
| `FedAvg` | 69.36% | $2k, 137.3GB$ | 49.58% | $30k, 608GB$ | 85.73% | $6k, 272GB$ |
| *biased* `FedAvg` | 83.9% | $2k, 130.92GB$ | 56.14% | $6k, 123.1GB$ | 85.82% | $6k, 272GB$ |
| Liang et al. (2020) | 83.16% | $2k, 129.93GB$ | 41.09% | $6k, 116.7GB$ | 86.75% | $6k, 271, 8GB$ |
| `FedMix` K=2 | 72.25% | $2k, 261.4GB$ | 48.29% | $30k, 1130GB$ | 84.02% | $6k, 544.4GB$ |
| `FedMix` K=4 | 79.88% | $2k, 522.5GB$ | 48.55% | $24k, 1972GB$ | 86.11% | $6k, 1089GB$ |
| `FedMix` K=10 | 80.00% | $2k, 1304.1GB$ | 52.59% | $6k, 1130GB$ | 87.21% | $6k, 2724GB$ |

from a different probability distribution over 8 possible rotation angles to rotate a digit with. Each client's distribution is drawn from $\text{Dir}(\alpha = 1.0)$. At test time, each data-point is randomly rotated according to the client's distribution. Additionally, we create a dataset where instead of uniform sampling of labels, we replicate the non-i.i.d label skew described for Cifar10 above and combine it with the rotation non-i.i.d-ness. We compare `FedMix` where $q$ is conditioned on $y$ or on the degrees of rotation for a data-point against baselines. Figures 5 and 6 show that *biased* `FedAvg` can improve in the presence of label skew but collapses to standard `FedAvg` for rotation non-i.i.d-ness. `FedMix` on the other hand has an advantage in both cases. Furthermore, in presence of both sources of non-i.i.d-ness, `FedMix` with $q(z|y)$ can outperform *biased* `FedAvg`. In the presence of only rotation (Figure 5), conditioning `FedMix` on the true source of non-i.i.d-ness is advantageous.

### 4.3 LABEL PERMUTATIONS

Apart from non-i.i.d-ness in $p(y)$ and $p(x)$, we can expect the mapping $p(y|x)$ itself to be different between clients. We replicate the experimental setup of (Sattler et al., 2019) with 20 clients for Cifar10, $C = 1.0$ and $E = 3$, albeit with LeNet-5. Each client is randomly assigned one of 4 different permutations of its labels, determining the cluster assignment $q(z|s)$ that `FedMix` has to learn. Although `FedMix` is designed to distinguish different regions of the input space, we show that it can perform user clustering. The gating function $p(z|x, s)$ learns to correctly identify, for each datapoint, the expert corresponding to the permutation of $s$, thus recovering the original clustering; Figure 7 illustrates this effect (please note that the ordering of columns is arbitrary). It should be mentioned, that `FedMix` converges very quickly compared to the results in (Sattler et al., 2019), as this result was obtained after only 10 rounds. After this point, `FedMix` essentially trains 4 independent models on the respective i.i.d subsets of the data that have the same label permutation. Appendix B contains additional discussions for $K \neq 4$ and experimental details.

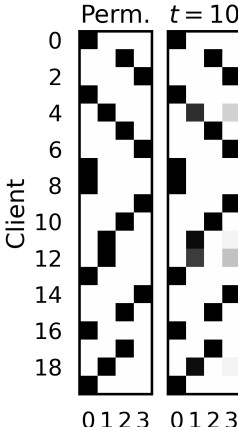

Figure 7: Ground truth and $q(z|s)$ after 10 rounds. Greyscale represents probabilities (white: 0; black: 1)

## 5 DISCUSSION

With `FedMix` we have introduced a federated learning algorithm that explicitly takes the non-i.i.d characteristics of a federated dataset into account. Clients can learn to align specialized experts on sub-regions of the data space and achieve higher performance compared to `FedAvg`, especially in situations where the source of the non-i.i.d nature is known. This assumption is very strong in real-world federated scenarios and we expect a more flexible alignment process than a global $q(z|y)$ to be the most interesting avenue for future research. We show encouraging results on experiments with non-i.i.d-ness in $p(y|x)$ (permutation) and $p(x)$ (rotation). In the future, we will explore ways to perform automatic selection of $K$ as well as automatic selection of architecture elements to share between experts, trading-off gradient alignment and communication budgets.

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

## A    EXPERIMENTAL SETUP

For Cifar10, we replicate the federated data split of Hsu et al. (2019). The dataset is split across 100 clients, whose data-points are drawn according to their label from a Dir($\alpha = 1.0$) distribution without replacement. For the base model, we use a LeNet-5 architecture (LeCun et al., 1998). We use a SGD optimizer with a learning rate of 0.05 and a batch size $B = 64$ locally and the Adam (Kingma & Ba, 2014) optimizer with its default hyperparameters at the server by interpreting the difference of the local from the global model as a gradient (Reddi et al., 2020). We sample 10 clients without replacement on each round (but with replacement across rounds) and train for $E = 1$ local epochs. For Cifar100 we replicate the data split of Reddi et al. (2020). The dataset is split into 500 clients by using a hierarchical model over the coarse and fine labels, with the same hyperparameters as the ones provided by Reddi et al. (2020). The other hyperparameters are the same as Cifar10 with the exception of the batch size, where we use $B = 20$, as well as the architecture, where we use a ResNet-20 with group normalization (Wu & He, 2018) layers instead of batch normalization (Ioffe & Szegedy, 2015). We augment the data by random cropping from a 4 pixel padded image and horizontal flipping. Finally, for the Femnist dataset, we similarly followed the setup of Reddi et al. (2020) with the same LeNet-5 architecture and hyperparameters of Cifar10 with the exception of the batch size where we used $B = 20$. $h_k(\mathbf{x})$ is defined as the input to an expert's output layer.

## B    PERMUTATION EXPERIMENTS

For the experiments in Section 4.3, we train `FedMix` with $K = 4$ and a base LeNet-5 model. The optimization hyperparameters are chosen identically to what is described in A, except for the optimization of $q(z|s)$. The local objective in this case is

$$L_s := \sum_{i=1}^{N_s} \sum_{z=1}^{K} \left[ q_\pi(z|s) \log p(y_{s,i}, z|x_{s,i}) - \beta q_\pi(z|s) \log q_\pi(z|s) \right]. \tag{10}$$

In order to speed up convergence, we found it beneficial to not perform gradient ascent and instead solve it directly using Lagrange multiplies, *i.e.*, $\nabla_{\pi,\lambda} L_s(D_s, \mathbf{w}, q_\pi) + \lambda(\sum_{k=1}^{K} \pi_k - 1) = \mathbf{0}$.

The solution for $\pi_k = q(z = k|s)$ at client $s$ takes the form of

$$\pi_k = \left( \prod_{i=1}^{N_s} p(y_{s,i}, z = k|x_{s,i})^{\frac{1}{N_s\beta}} \right) \Bigg/ \left( \sum_{z=1}^{K} \prod_{i=1}^{N_s} p(y_{s,i}, z = k|x_{s,i})^{\frac{1}{N_s\beta}} \right). \tag{11}$$

For these permutation experiments, `FedMix` is tasked to recover a cluster-assignment, as opposed to a partitioning of the input space. We have found it unnecessary to change $\beta$ from its default value of 1. For an approximation of $\pi_k$ with a mini-batch of size $M$, we approximate

$$\log \pi_k = \frac{1}{N_s} \sum_{i=1}^{N_s} \log p(y_{s,i}, z = k|x_{s,i}) - \log \sum_{z=1}^{K} \exp \frac{1}{N_s} \sum_{i=1}^{N_s} \log p(y_{s,i}, z = k|x_{s,i}) \tag{12}$$

$$\approx \frac{1}{M} \sum_{i=1}^{M} \log p(y_{s,i}, z = k|x_{s,i}) - \log \sum_{z=1}^{K} \exp \frac{1}{M} \sum_{i=1}^{M} \log p(y_{s,i}, z = k|x_{s,i}) \tag{13}$$

and perform dampening $\pi^t = \alpha \cdot \pi^{t-1} + (1 - \alpha) \cdot \pi$. We found our initial pick of $\alpha = 0.75$ to perform well.

### B.1    MODIFYING $K$

In Section 4.3, we choose $K = 4$ a-priori as the number of experts that fits the data generating process with 4 clusters. in Figure 8, we plot $q(z|y)$ in case we miss-specify $K$ and choose $K = 3$ and $K = 5$. In this plot, we can consider $K = 4$ as the ground truth for the correct cluster assignment. Please note that ordering of the columns is arbitrary and can be different across the three experiments. Comparing $K = 4$ with $K = 5$, we see that `FedMix` assigns two experts to the same cluster, while still correctly disentangling all clients. For $K = 3$, `FedMix` correctly identifies two clusters while being forced to mix together the other two clusters into one expert.

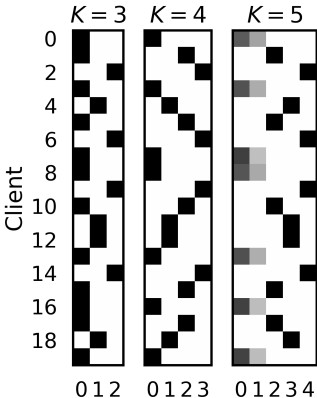

Figure 8: $q(z|s)$ at convergence for different number of experts $K$. Greyscale corresponds to probabilities (white: 0; black: 1)

## C  PRIVACY IMPLICATIONS

Privacy is one of the key motivations for research and deployment of Federated Learning. Even though privacy is not a focus of this paper, we briefly discuss some implications of making explicit use of $q(y|s)$ in FedMix. The update rule described in equation 8 requires access to the marginal $q(z|s) = \sum_y p(y|s)q_\phi(z|y)$ at the server. At the same time, the server has access to the parameters $\phi$ that were used in computing $p(z|s)$ before being sent to the server. Therefore, in principle, it could solve $q(z|s) = \sum_y p(y|s)q_\phi(z|y)$ with respect to $p(y|s)$ and thus obtain the marginal label distribution at the client. In practice this is not as straightforward to do as (a) the probability matrix $q_\phi(z|y)$ is not always invertible and (b) solutions that use the pseudo-inverse, empirically, are not very accurate in capturing the entire distribution. With the additional constraints that the marginal needs to sum to one contains only positive elements and that $N_s \cdot p(y|s) \in \mathbb{Z}$, in some cases, a reconstruction can become possible. As the number of classes exceeds the number of experts, this reconstruction becomes more unlikely. We leave a thorough characterization of these properties to future work.

Irrespective of FedMix, we want to shine light on the possibility to reconstruct $p(y|s)$ at the server-side in standard FedAvg. Assume a randomly initialized model being sent to a client $s$, where the client performs a single full data set update step on the output layer's bias vector $b_s$. Assuming a softmax cross-entropy loss $L_s$, the average gradient with respect to a the k-th entry $b_k$ takes the form of

$$\frac{\partial L_s}{\partial b_k} = \frac{1}{N_s} \sum_{i=1}^{N_s} 1[y_{i,k} = y_{i,\text{true}}] - \pi_{i,k}, \qquad (14)$$

where 1 is the indicator function and $\pi_{i,k}$ is the softmax probability of class $k$ of datapoint $i$. With a randomly initialized model, these softmax probabilities can be assumed to be uniform, leading to an average gradient of

$$\frac{\partial L_s}{\partial b_k} = \frac{N_{s,k}}{N_s} - \frac{1}{N_c} = p(y|s) - \frac{1}{N_c}, \qquad (15)$$

where $N_c$ is the number of classes. Upon sending the updated bias vector $b_s = b - \alpha \frac{\partial L_s}{\partial b_k}$ to the server, it can easily reconstruct the marginal label distribution.

Figure 9 shows, for every client in our Cifar10 setup (Appendix A), in red the true marginal $p(y|s)$ and in blue the reconstructed marginal based on (the same) randomly initialized model being sent to each client. Clearly, we have high congruence. In Figure 10, we investigate the more realistic setup of $E = 1$ with mini-batch stochastic gradient descent at the client level. In order to avoid reconstructing the multi-step update, we simply normalize the difference $(b - b_s)$ and interpret it

as marginal label distribution. We see that multiple updates (on average: 8) reduce the congruence between the true and reconstructed marginal, however the information leakage is still remarkable.

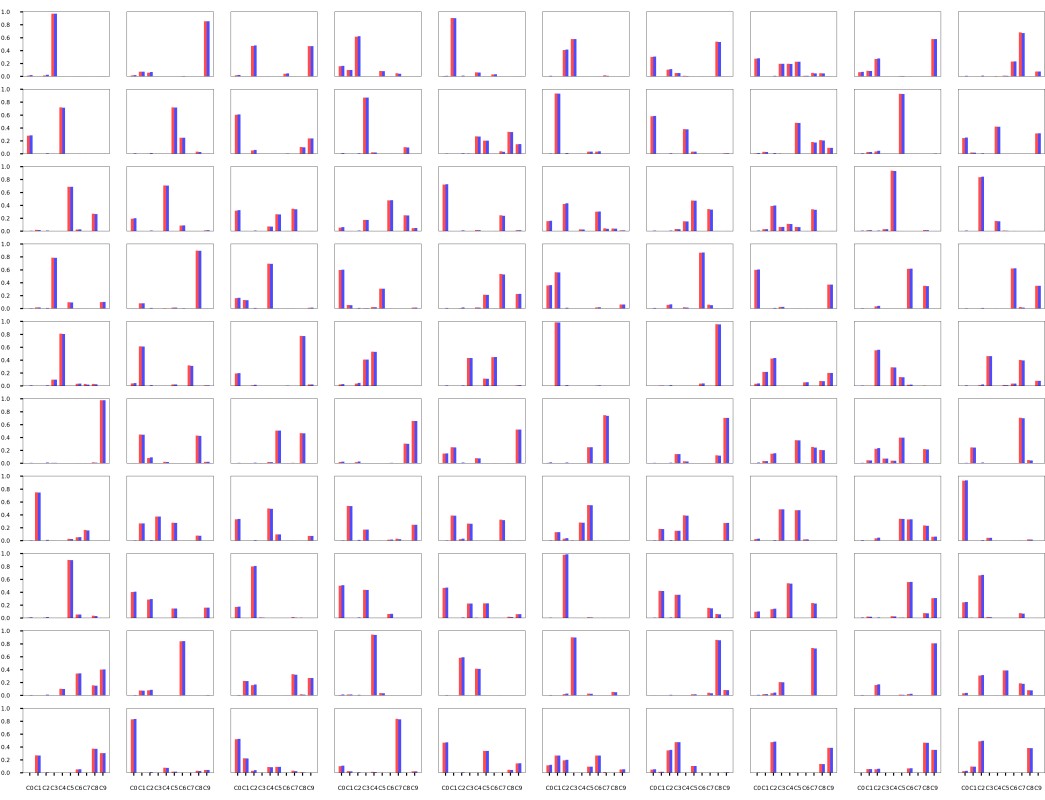

Figure 9: Histogram representation of $p(y|s)$ as well as its server-side reconstruction for every client $s$ for the Cifar10 setup described in Appendix A. The x-axis per sub-plot enumerates the 10 classes. For every class $c$, the left bar in red represents $p(y = c|s)$ and the right bar in blue represents its reconstruction. Each client performed a single full data-set update step.

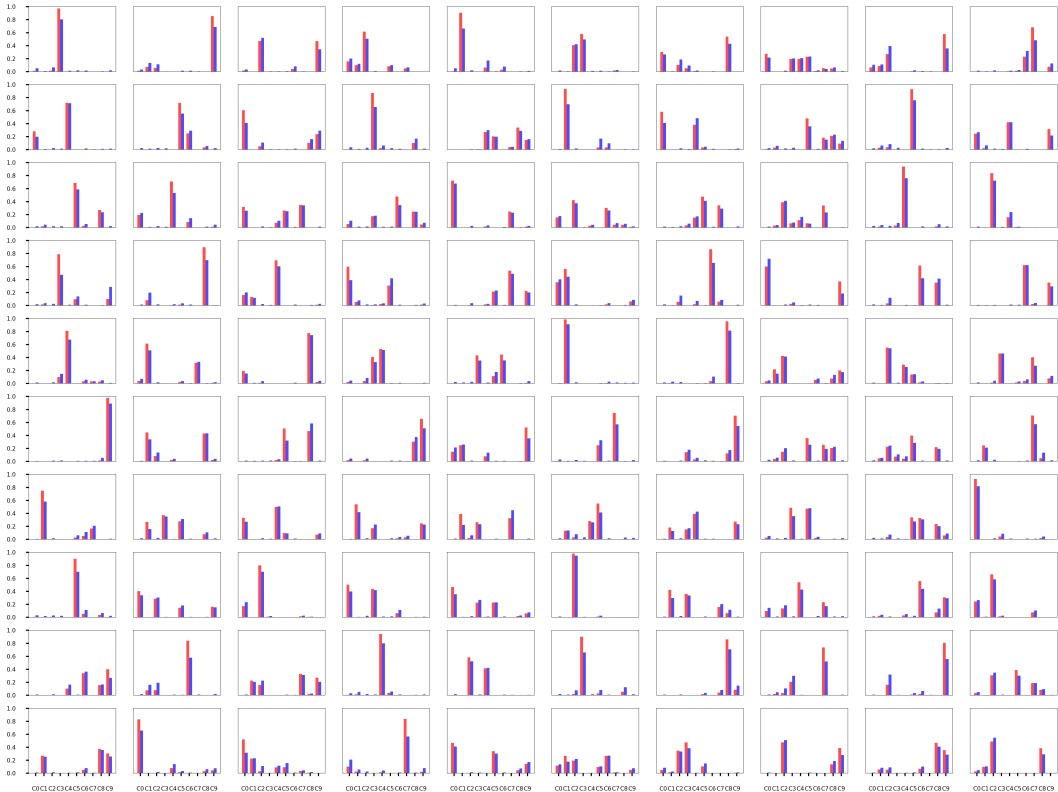

Figure 10: Histogram representation of $p(y|s)$ as well as its server-side reconstruction for every client $s$ for the Cifar10 setup described in Appendix A. The x-axis per sub-plot enumerates the 10 classes. For every class $c$, the left bar in red represents $p(y = c|s)$ and the right bar in blue represents its reconstruction. Each client performed multiple mini-batch update steps (on average 8).

## D    GRADIENT DIVERGENCE

We aim to track the divergence of updates for a subset $S'$ of shards at the server at time step $t$ for `FedMix` and `FedAvg`. Therefore we define a metric inspired by Kim et al. (2020); Sattler et al. (2019) to define divergence of gradients $\Delta_{k,i}^t = \omega_k^t - \omega_{k,i}^{t+1}$ for some subset $\omega_k$ of the parameters $\mathbf{w}_k$ of expert $k$ as

$$\text{GD}(\Delta_k^t) = \sum_{i=1}^{S'} \sum_{j=1}^{S'} p(s = i|z = k)p(s = j|z = k) \cdot 0.5 \cdot \left(1 - \frac{\Delta_{k,i}^t \cdot \Delta_{k,j}^t}{||\Delta_{k,j}^t|| \cdot ||\Delta_{k,i}^t||}\right). \quad (16)$$

For `FedAvg`, the above metric collapses to

$$\text{GD}(\Delta^t) = \sum_{i=1}^{S'} \sum_{j=1}^{S'} p(s = i)p(s = j) \cdot 0.5 \cdot \left(1 - \frac{\Delta_i^t \cdot \Delta_j^t}{||\Delta_j^t|| \cdot ||\Delta_i^t||}\right). \quad (17)$$

In Figure 1 in the main text, we plot the sum of $\text{GD}(\Delta_k^t)$ across all parameters $\omega_k$ (*i.e.*, convolutional kernels, weights and biases) of the LeNet-5 experts in comparison to $\omega$ for `FedAvg`.

# E  ABLATION STUDIES

We investigate the characteristics of `FedMix` by varying the number of experts $K$ for Cifar10, as well as by selecting different levels $\eta$ for pruning experts. Figures 11a and 11c show learning curves of several values of $K$. We can see that a higher number of experts has the potential to achieve higher accuracies at the cost of more required communication. With increasing $K$, the modeling task for a single expert becomes progressively easier and the challenge is moved towards identifying the right expert for a given data-point. This could subsequently lead to overfitting in the federated setting if not for the shared feature extractor $h(\mathbf{x})$ (Eq. equation 9). In case we choose $K$ to be equal to the number of classes, we observe that `FedMix` assigns one cluster to each class and the $p(z|\mathbf{x}, s)$ is reminiscent of Liang et al. (2020). In case we further increase $K$, we observe marginally improved performance by multiple experts being assigned to the same label.

Figures 11b and 11d depict the influence of different pruning levels $\eta$ on the communication efficiency and final performance `FedMix` with $K = 4$. We can see that higher levels of pruning do reduce communication significantly, however they cause some loss of performance.

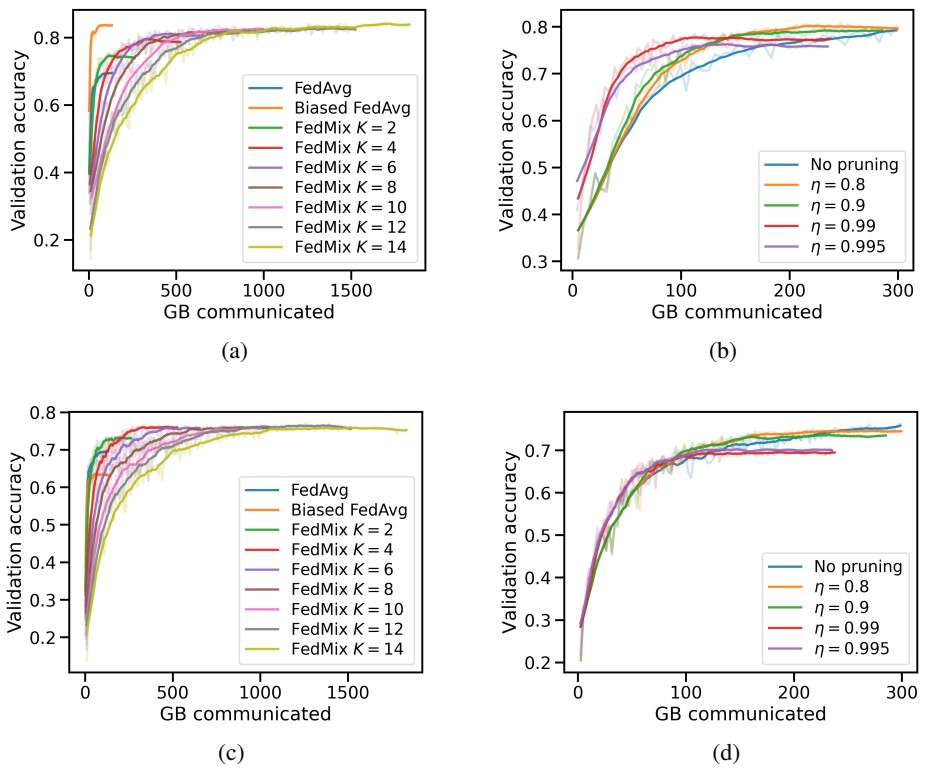

Figure 11: Ablation studies on the effect of $K$ on the average local accuracy (**a**) and new shard accuracy (**c**) as well as on the influence on local (**b**) and new shard (**d**) accuracy of the threshold for pruning experts before communication for $K = 4$.

# F  NEW SHARD INFERENCE

In the main text, we discussed performance of the individual algorithms when evaluating new data on a shard that took part in training and therefore has access to trained local model parameters. This is the case for all considered methods except `FedAvg`, for which there exist no localized parameters. For inference on a new shard, we use the sum of all local validation sets as a proxy. Liang et al. (2020) propose to ensemble the representations of the local feature extractors across clients before evaluating the global part of the network. We find this approach to work quite poorly in practice. Presumably pretraining the local feature extractors with `FedAvg` ameliorates this behaviour. In

*biased* `FedAvg`, we ensemble the individual local biases across clients to receive a single global bias. For `FedMix`, we marginalize the local gating predictions to achieve a global gating prediction $p(z|\mathbf{x}^*) = \sum_{s=1}^{S} p_{\theta_s}(z|\mathbf{x}^*, s)p(s)$. Predictions can then be made by marginalizing across experts using this global gating function: $p(y|\mathbf{x}^*) = \sum_{z=1}^{K} p_{\mathbf{w}_k}(y|\mathbf{x}^*, z)p(z|\mathbf{x}^*)$.

Interestingly, Cifar10 and Cifar100 display different behaviours when comparing `FedMix` to the baselines. For Cifar10, *biased* `FedAvg` performs much worse compared to `FedAvg` while `FedMix` performs well. For Cifar100, *biased* `FedAvg` is indistinguishable from `FedAvg`, outperforming `FedMix`. We leave the investigation of these discrepancies to future work.

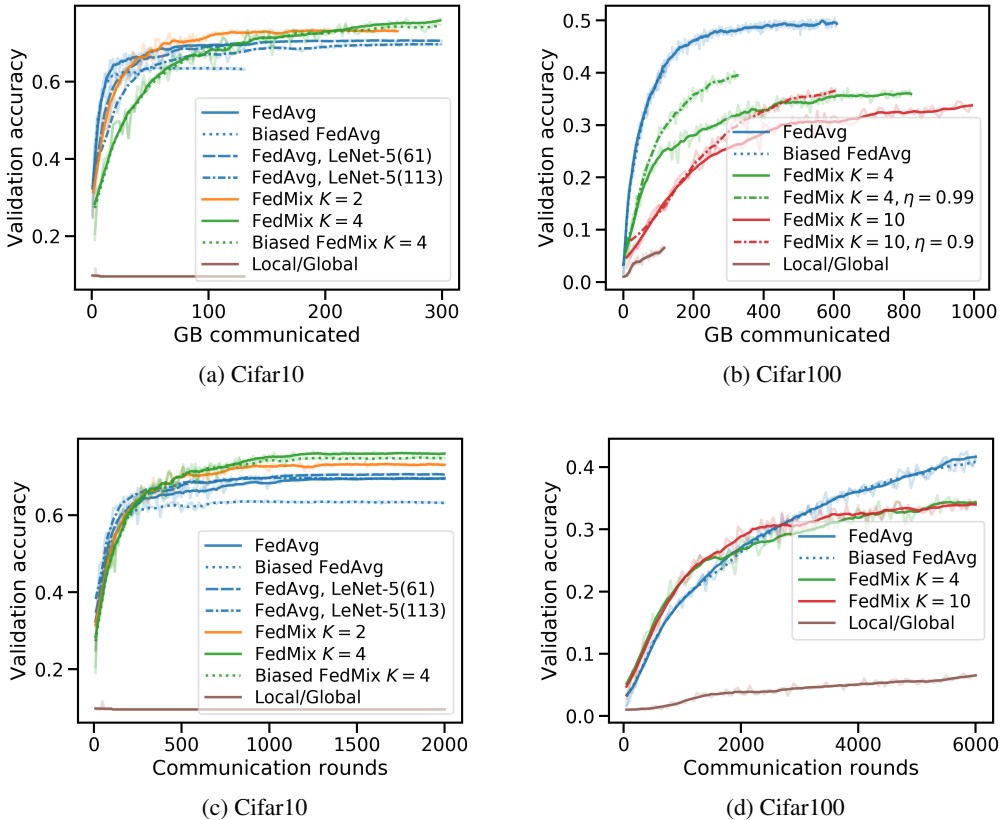

Figure 12: Accuracy on a new client (y-axis) as a function of the amount of GB communicated (top row) and as a function of communication rounds (bottom row). Cifar 10 models are trained on the standard 45k training split. x-axes have been truncated for improved visibility. Best viewed in color.

As an alternative to marginalizing the local gate predictions using $p(s)$, investigating the graphical model in Figure 2 reveals the possibility for marginalization with $p(s|\mathbf{x}^*)$:

$$p(z|\mathbf{x}^*) = \sum_{s=1}^{S} p_{\theta_s}(z|\mathbf{x}^*, s)p(s|\mathbf{x}^*), \quad p(s|\mathbf{x}^*) \propto p(\mathbf{x}^*|s)p(s). \tag{18}$$

It is for this third evaluation case that training local generative models $p(\mathbf{x}|s)$ for each client becomes interesting, as they allow to compute the responsibilities $p(s|\mathbf{x}^*)$ at test time. In practice, however, the success of this approach depends heavily on the correctness of $p(s|\mathbf{x}^*)$, which in turn depends on the ability of the local generative models $p(\mathbf{x}|s)$ to assign high probability to data that resembles $\mathcal{D}_s$ and low probability to out-of-distribution data. Training and calibrating generative models for this task is in itself an active area of research (Nalisnick et al., 2018) and investigating how clients in a federated setting might exchange information to facilitate this process is not yet explored. We

therefore leave a thorough evaluation of this case to future work and only present here a MNIST (LeCun et al., 2010) experiment.

We train `FedMix` with $K = 4$ and experts of two hidden layer ReLU MLP with 200 hidden units on MNIST. We split the dataset into $S = 100$ clients according to the procedure described in Liang et al. (2020). `FedMix` achieves 97.7% average validation accuracy compared to 97.0% with `FedAvg` after 600 communication steps. Independently for each client, we train a small variational autoencoder with a 32-dimensional latent space using a two-layer MLP with 512 and 256 hidden units respectively as encoder and mirrored decoder structure. We optimize the VAEs using Adam with standard hyper parameters, $B = 10$ and perform early stopping on the local validation sets after no improvement for three epochs. After training, each client communicates their VAE to the server, where we evaluate $p(s|\mathbf{x}^*)$ to marginalize over the local gating functions according to $p(z|\mathbf{x}^*) = \sum_{s=1}^{S} p(z|\mathbf{x}^*, s)p(s|\mathbf{x}^*)$. With this procedure, `FedMix` achieves 95.9% test set accuracy, compared to `FedAvg` with 96.9%. Marginalizing with $p(s)$ instead of $p(s|\mathbf{x}^*)$ achieves 96.63%, showing the limitation of the approach in that any error in $p(s|\mathbf{x}^*)$ propagates into the expert assignment. Reliable out-of-distribution detection capabilities in the individual estimators for $p(\mathbf{x}|s)$ are therefore necessary.

## G   LOCAL FINE-TUNING

In the main text we have compared algorithms by evaluating local data-sets on the server-provided global parameters. For *biased* `FedAvg`, Local/Global and `FedMix` those parameters were made complete with the client-specific local parameters. In Appendix F we have discussed the performance of `FedMix` when evaluating on a new shard. Here, we discuss the evaluation when using locally fine-tuned models. Figure 13 shows the learning curves of Cifar10 when evaluating the validation set on parameters just before communication to the server. This corresponds to local fine-tuning for $E = 1$ epochs on the local data-set. This form of model personalization is an option for all federated learning methods. Due to the relatively small size of the local data-sets, this form of personalization can cause significant levels of overfitting, as is evident from Figure 13. Since regularization with respect to this form of personalization adds another level of complexity to the experimental design and hyperparameter tuning, we leave a thorough evaluation to future work.

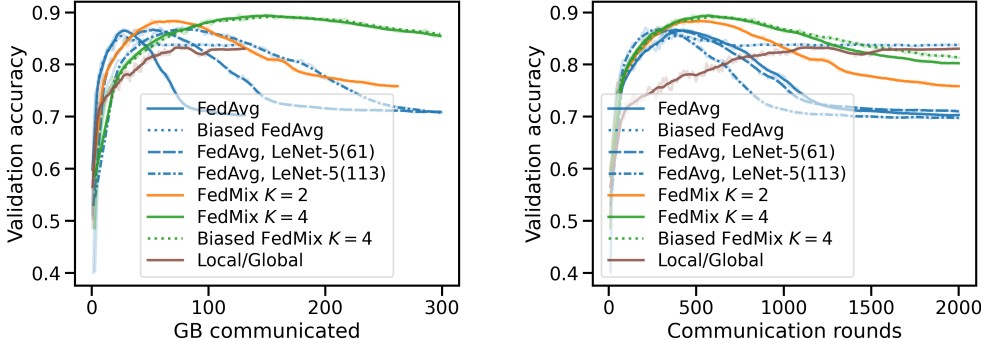

Figure 13: Average accuracy of models evaluated after $E = 1$ local epochs of fine-tuning

## H    SPECIALIZATION

In Figure 14a we see how $q_\phi(z|y)$ changes over time. The entropy of $q_\phi(z|y = c)$, i.e. the weight with which each expert is assigned to a specific label $c$ decreases over time until each label is assigned exactly one expert with probability 1. The original MoE formulation, however, quickly collapses to a single expert, as can be see in Figure 14b. In the federated learning scenario, this corresponds to performing standard `FedAvg` on the single surviving expert.

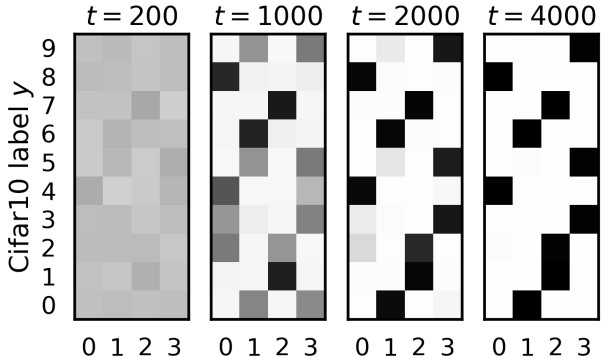

(a) Visualization of $q_\phi(z|y)$ at different communication rounds $t$ for `FedMix` with $K = 4$ on Cifar10. Greyscale corresponds to probabilities; white corresponds to zero and black corresponds to one. Probabilities sum to one across experts (horizontally).

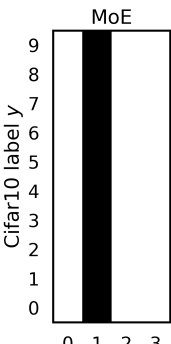

(b) Visualization of $\sum_{i=1}^{N} p(z|x_i)$ for the original mixture of experts formulation (equation 3 with $S = 1$) after the first epoch of training. The mixture collapses to a single expert.

Figure 14: Expert specialisation

## I    FEDMIX WITH EXPERT PRUNING

The `FedMix` algorithm (Algorithm 1) can be extended to ignore the communication of experts between server and client if, for this given client, that particular expert is not updated during training. If $q_\phi(z = k|s) = E_{p(y|s)}[q_\phi(z = k|y)] = 0$ for an expert $k$ on client $s$, then parameters of that expert observe no gradient and the expert is effectively pruned from the local library of available experts. Since the parameterization of $q_\phi(z|s)$ does not allow for exact values of zero, we introduce $\eta$, such that if $q_\phi(z = k|s) < \eta/K$, we consider expert $k$ to be pruned. Since the value of $q_\phi(z = k|s)$ during local optimization at the client is subject to change, it is possible that $q_\phi(z = k|s)$ briefly lies above the threshold even though it had just been pruned away. Therefore the server considers expert $k$ pruned for a given client $s'$ only if $q_\phi(z = k|s = s') < 0.9 \cdot \eta/K$. Algorithm 2 details this approach. Note that we write $p(z|s) = \mathbb{E}_{\mathbf{x} \sim D_s}[p_{\theta_s}(z|\mathbf{x}, s)]$, i.e. the true marginal, however in practice we make use of the cheap to compute marginal approximation $q_\phi(z|s) = \mathbb{E}_{y \sim D_s}[q_\phi(z|y)]$.

**Algorithm 2** The `FedMix` algorithm. $\alpha, \beta$ are the client and server learning rates respectively, $\eta \in [0, 1]$ is the pruning threshold

---

**function** SERVER SIDE
    Initialize $\phi$ and $K$ vectors $\mathbf{W} = [\mathbf{w}_1, \ldots, \mathbf{w}_K]$
    **for** round $t$ in $1, \ldots T$ **do**
        $S' \leftarrow$ random subset of the clients
        Initialize $\Delta_{\mathbf{W}}^t = \mathbf{0}, \Delta_{\phi}^t = \mathbf{0}$
        Initialize $p(z|s) = 1/K \forall s$
        **for** $s$ in $S'$ **do**
            $\mathbf{W}' \leftarrow [\mathbf{w}_k \,|\, p(z = k|s) \geq 0.9 \cdot \eta/K]$
            $\mathbf{W}_s^t, \phi_s^t, p(z|s) \leftarrow$ CLIENT SIDE$(s, \phi, \mathbf{W}')$
            Store $p(z|s)$
        **end for**
        $p(s|z) \leftarrow p(z|s)p(s)/\sum_{s \in S'} p(z|s)p(s)$

        **for** $s$ in $S'$ **do**
            $\Delta_{\mathbf{w}_k}^t \mathrel{+}= p(s|z = k)(\mathbf{w}_k^{t-1} - \mathbf{w}_{s,k}^t) \;\; \forall k$
            $\Delta_{\phi}^t \mathrel{+}= \frac{N_s}{N_{S'}}(\phi^{t-1} - \phi_s^t)$
        **end for**
        $\Delta_{\phi}^t \mathrel{-}= \nabla_{\phi} H(\sum_c q_{\phi}(z|y{=}c)p(y{=}c))$
        $\mathbf{w}_{1:K}^{t+1} \leftarrow$ ADAM$(\Delta_{\mathbf{w}_{1:K}}^t, \beta)$
        $\phi^{t+1} \leftarrow$ ADAM$(\Delta_{\phi}^t, \beta)$
    **end for**
**end function**

**function** CLIENT SIDE$(s, \phi, \mathbf{W})$
    Get local parameters $\theta_s$
    **for** epoch $e$ in $1, \ldots, E$ **do**
        **for** batch $b \in B$ **do**
            $q_{\phi}'(z|s), q_{\phi}(z|s) \leftarrow \mathbb{E}_{y \sim \mathcal{D}_s}[q_{\phi}(z|y)]$
            $q_{\phi}'(z = k|s) = 0$ if $q_{\phi}(z = k|s) < \eta/K$
            $q_{\phi}'(z|s) \leftarrow q_{\phi}'(z|s)/\sum_k q_{\phi}'(z = k|s)$
            $q_{\phi}'(z|y) \leftarrow q_{\phi}(z|y)$
            $q_{\phi}'(z = k|y) = 0$ if $q_{\phi}(z = k|s) < \eta/K$
            $q_{\phi}'(z|y) \leftarrow q_{\phi}'(z|y)/\sum_k q_{\phi}'(z = k|y)$
            $L_s \leftarrow \mathbb{E}_{q_{\phi}'(z|y_b)}[\log p_{\mathbf{w}_z}(y_b|\mathbf{x}_b, z)] +$
                    $\mathbb{E}_{q_{\phi}(z|y_b)}[\log p_{\theta_s}(z|\mathbf{x}_b, s)]$
            $\phi \mathrel{+}= \alpha \nabla_{\phi} L_s$
            $\mathbf{W} \mathrel{+}= \alpha \nabla_{\mathbf{W}} L_s$
            $\theta_s \mathrel{+}= \alpha \nabla_{\theta_s} L_s$
        **end for**
    **end for**
    $q(z|s) \leftarrow \mathbb{E}_{y \sim \mathcal{D}_s}[q_{\phi}(z|y)]$
    $\mathbf{W}' \leftarrow [\mathbf{w}_k | q(z = k|s) \geq \eta/K]$
    $q'(z|s) \leftarrow q(z|s)$
    $q'(z = k|s) = 0$ if $q(z = k|s) < \eta/K$
    $q'(z|s) \leftarrow q'(z|s)/\sum_k q'(z = k|s)$       ▷ Renormalization
    **return** $\mathbf{W}'; \phi; q'(z|s)$
**end function**

---

