# OpenReview forum: "Federated Mixture of Experts"
_ICLR.cc/2021/Conference — Reject_

### Official Review · AnonReviewer3 · 2020-10-29
**Interesting approach, more in-depth experimental evaluation and theoretical insights needed**

**Rating:** 5
**Confidence:** 4

**Review:**

The paper proposes a novel algorithm, which is a federated form on mixture of experts, called Federated Mixture of Experts (FedMix). In FedMix, an ensemble of specialized models is trained instead of a single global model. This strikes a compromise between training a single global model and one model per client. A gating mechanism is employed, to choose an expert model that is responsible for a given data point, thus aligning the gradient updates across experts and alleviating the consequences of non-i.i.d. data.

Pros:
- the paper is well written and easy to follow
- it tackles an important problem, namely how to cope with non-iid data in FL
- experiments are performed on different datasets

Cons:
- A major drawback of the paper is the lack of  in-depth experimental evaluation and theoretical insights. On Cifar-10 and Cifar-100 the proposed method performs worse than related state-of-the-art methods. On the rotated MNIST experiments in 4.2 it performs better, however, this section is very short and the insights from the presented results are limited.

- The STC approach presented in "Robust and Communication-Efficient Federated Learning from Non-IID Data" performs even better than biased FedAve on Cifar-10. It may well be that it also performs well in the rotated MNIST experiments. Anyways I am not convinced that the proposed method is superior to state-of-the-art techniques in non-iid settings (on Cifar-10 and -100 it is clearly not).

- The paper does not provide any new theoretical insights, it only reports empirical results. For instance, can you give guarantees that the proposed approach will result in K specialised models if the client data was generated from K different distributions (see  proof in Sattler et al. 2020).

- The authors mention the conceptual similarity of the proposed approach to Clustered Federated Learning (Sattler et al. 2020, Briggs et al., 2020), that allow to jointly train specialised models. Why do you not compare the proposed approach to these methods? I assume that Clustered FL will solve the rotated MNIST problem (although it may be worse in terms of communication-efficiency). More insightful experiments would be great.

Overall, I am not convinced by the proposed method and be the reported results.

---

> ### Author Response · Authors · 2020-11-19
> **Response to AnonReviewer3**
>
> Thank you for your assessment of our paper. We are glad you found the paper easy to follow and its scope as relevant.
>
> The reviewer mentions our experimental evaluation as a major drawback of our paper. Similarly to the other reviewers’ comments, we would like to point out that our proposed biased FedAvg baseline is specialized to the label-skew setting, whereas FedMix generalizes to other sources of non-iid-ness. We showcase that in the case of rotated MNIST as well as additional experiments based on your suggestion of CFL [Sattler et. al 2019]. You are right that FedMix is clearly not superior on the label-skew experiments – however we would expect any specialized method to outperform a more general one.
>
> Please note that the STC paper experiments with a modified VGG architecture, whereas we experiment with LeNet-5 on Cifar10. The results are thus not comparable.
> The method of Clustered Federated Learning (CFL) is indeed similar to FedMix but crucially clusters users, whereas FedMix assigns portions of the input-space to different experts. We have taken your critique as an opportunity to show empirically that FedMix can recover the correct permutation in the setting that is discussed in the CFL paper.
>
> We hope that the additional empirical observations will convince you about the usefulness of FedMix and you consider raising your score.

---

### Official Review · AnonReviewer4 · 2020-10-30
**The idea is interesting but improvements are expected**

**Rating:** 4
**Confidence:** 4

**Review:**

Strength:
* The idea of training an ensemble of specialized models in FL is interesting, although not novel.
*  The evaluation is solid. I would be more impressed if the authors could also evaluate some SOTA language models.

Weakness:
*  The technical contribution is limited and the novelty is not sufficient. I am not against "A+B" research if it does bring drastic practical improvement. However, the proposed method is worse than some prior works on CIFAR 10 even using ten-time bandwidth (correct me if I interpret the results wrongly).
*  There are many other works with similar technical consideration although their intention might be different. Please check the line of works about personalized FL and compare with them if possible.
*  The writing is not clear given that the idea is actually simple.

---

> ### Author Response · Authors · 2020-11-19
> **Response to AnonReviewer4**
>
> We thank the reviewer for their assessment.
> Please let us know which references for mixture models in FL we are missing – we will gladly improve our paper by including further references. If possible, we will compare against those methods, given that the devil is in the details of the experimental setup.
>
> Language models require some considerations for how to construct the gating mechanism and we leave such models to future work. Given proper construction, we expect FedMix to work for recurrent or transformer-based models.
>
> We agree that FedMix in its naïve version requires K times more bandwidth compared to some specialized approaches. FedMix does not require label-skew, as does, for example, our biased FedAvg baseline. We therefore believe that the federated mixture of experts approach can serve as a more general approach to non-iid-ness. To that extent, we have included a new discussion about the $p(y|x)$ non-i.i.d-ness setting and show that FedMix can recover it perfectly.
>
> Please let us know where we can improve upon the clarity of our work.

---

### Official Review · AnonReviewer1 · 2020-11-05
**A (personalized) mixture of experts, but with heavy communication and privacy cost**

**Rating:** 4
**Confidence:** 4

**Review:**

The authors propose using an ensemble of experts, but where the mixing proportions depend on both the data and the client (providing the personalization in the method). The proposed method is very interesting, but there are several issues which need to be addressed.

The method requires significantly higher communication cost to train, since multiple versions of the model parameters need to be communicated, as well as parameters for other models used for the mixing. The authors propose suggestions for reducing the number of models that need to be communicated, but they rely on very special behaviour of the different models (namely that they separate well so that clients do not actually need all of them).

The clients need to transmit mixing parameters $q(z|s)$ as well as the $\phi$ which generate them. Given access to $N_s$ and the discrete nature of $y$, this would allow the server to reconstruct the label distribution of each user's data, which is a huge reduction in the privacy achieved by federated learning.

The algorithm is quite complicated, and appears fairly fragile, as is demonstrated by the different functions for $h_k(x)$ required for just 3 data sets. Selection of tuning parameters in federated learning is a challenging and task, and having one which is a function makes the method quite challenging to use in practice.

Experimental results are not very compelling, with a simple personalized bias term outperforming the proposed model in terms of both number of communication rounds and total volume of data transmitted, and does not produce the privacy violation the proposed method produces. Although the simple personalized bias does not generalize beyond label distribution, there are many many similar personalization proposals for federated learning which do, and it is not clear whether the proposed method is competitive with those. Of course comparison with all existing methods is not feasible, but given the method must be entirely judged by empirical results, comparison with at least some should be done.

Overall it is not clear that the extra costs induced by the method are justified by the improvement in convergence speed, especially compared to more simple personalization methods which do not incur any of these extra costs (either in communication or privacy).

---

> ### Author Response · Authors · 2020-11-19
> **Response to AnonReviewer1**
>
> We thank the reviewer for their feedback. We are glad that you find it interesting and would like to address your concerns.
>
> You mention significant communication costs. We agree that FedMix in its naïve form requires K times the communication budget. We propose FedMix as a way to combat non-iid-ness subject to this additional cost. The additional costs of communicating $q(z|s)$ and $\phi$, however, are negligible: For Cifar10, for example, $\phi$ is of size $(10\times K)$ and $q(z|s)$ is just $1\times K$.
>
> We thank you for pointing out the potential privacy implications on inferring the label marginal $p(y|s)$ at the server-side. We have included, in the Appendix, a discussion on when a reconstruction of the marginal with FedMix is possible. Roughly, this is possible when the probability matrix q(z|y) is invertible, and that is generally only possible if $K \geq$ # classes. It should be noted that FedAvg can also suffer from a similar attack, as we can infer the marginal label distribution via the gradient of the bias of the output layer. We have included a discussion about this and empirical results in the Appendix as well.
>
> You mention the different design of the gating function across experiments. We have revised our Femnist experiments such that the architectural design of FedMix is now consistent across all our experiments.
>
> Your main critique of our experimental evaluation is that FedMix is being outperformed by other personalization approaches. We compare against the (limited) biased FedAvg and the Local/Global method. If you have further methods against which a comparison would convince you, we are glad to evaluate them. In our experience, the devil is often in the detail of the experimental setup (number of users, model, source of non-iid-ness, sampling rate of clients, number of local epochs etc.), so it is in general hard to make a paper-to-paper comparison without actually implementing the competing method in the same environment. If at all feasible within the rebuttal period, we will aim to compare if you provide us with concrete references.
>
> As we have mentioned in our comment to Reviewer2, the federated mixture of experts approach is generally applicable for many sources of non-iid-ness without being restricted to specific data-sets.
> We have added experiments on the setting where non-i.i.d-ness stems from $p(y|x)$, a setting where (biased) FedAvg fails and FedMix does not. Although FedMix might be out-performed by specific methods targeting a specific non-i.i.d.ness source, it presents a compelling option ‘across all data-sets’, so to say.
>
> We hope that with our argumentation about the privacy of FedMix, our additional experiments and clarifications, you will re-evaluate our contribution and consider to raise your score.

---

> > ### Comment · AnonReviewer1 · 2020-11-23
> > **Very interesting privacy exposition**
> >
> > Thank you for this additional privacy discussion for FedAvg, as well as for addressing the issue with FedMix. Is it not possible that even when the probability matrix is not invertable that you could still solve for the probabilities by using knowledge of the total number of points on the client, and the integer constraints on all the label distributions (i.e. us the fact that when each label proportion is multiplied by the number of points on the client it must still be an integer)?
> >
> > As for the issue with FedAvg, this is an interesting point that FedAvg is also vulnerable to a similar issue (at least during the first update). However it seems like this would be easier to solve (for example by not running full local epochs), whereas for FedMix these proportions are integral to the method.
> >
> > For comparison with personalization, there are many methods of personalization, and you need not test all of them but just having one comparison would be sufficient. The concern here is you can get most of the gains using another personalization method which does not have these same communication costs and privacy concerns.

---

> > > ### Author Response · Authors · 2020-11-24
> > > **Responds to your comment**
> > >
> > > Thank you for engaging with us once more! We agree that the integer constraints are additional prior knowledge that we hadn’t considered yet – however we fail to see how that solves the fundamental issue with invertibility. Maybe the reviewer has some additional pointers.
> > > For the privacy implications of FedAvg, we would assume a similar insight into the bias vector even if the client does not compute for a full epoch – that is assuming the client performs random sub-sampling of its dataset. On a more fundamental comparison with FedMix, we note that the same type of analysis is applicable to any type of FL algorithm that updates its bias vector with (S)GD locally and communicates it to the server. Our exposition was intended to point out that FedMix does not offer up additional information compared to what can be inferred with FedAvg in the first place.
> > >
> > > We would like to point out to the reviewer that we have a personalization-specific method, Local/Global [Liang et. al. 2020] in the paper. Additionally, we have added in our rebuttal the ‘personalization by fine-tuning’ results for all of our methods. We hope that those give you enough insight into the personalization angle of FedMix.

---

> > > > ### Comment · AnonReviewer1 · 2020-11-24
> > > > **Brute force**
> > > >
> > > > Unless both the number of points per client and the number of classes were large, would you not be able to brute force this? The number of possible solutions is $n^{k-1}$ and each check is a small amount of linear algebra.

---

> > > > > ### Author Response · Authors · 2020-11-24
> > > > > **Brute Force still underdetermined**
> > > > >
> > > > > A brute-force enumeration of all $p(y|s)$ is certainly possible under the assumption of discrete number of data-points per class. However since the system is still underdetermined, it will generally not be possible to identify which of those $p(y|s)$ is the *true* $p(y|s)$.
> > > > >
> > > > > Consider this concrete example (with arbitrary values).
> > > > > For a problem with $3$ classes, assume a client has the following number of data-points per class:
> > > > >
> > > > > $N_{y1} = 5$, $N_{y2} = 3$, $N_{y3}=2$, with a total of $10$ data-points.
> > > > > We assume $K=2$ experts, i.e $z \in${1,2} and a $q(z=1|y)$ as follows:
> > > > >
> > > > > $q(z = 1 | y=1) = 8/10$
> > > > >
> > > > > $q(z=1 | y=2) = 1/10$
> > > > >
> > > > > $q(z=1 | y=3) = 1/10$
> > > > >
> > > > > $q(z=2|y=c)$ is then equal to $1-q(z=1|y=c)$
> > > > >
> > > > > For this client, $q(z=1|s) = 45/100$ and $q(z=2|s) = 55/100$
> > > > >
> > > > > If you attempt solve the system of equations that says
> > > > >
> > > > > $q(z=1|s) = 0.8x + 0.1y + 0.1z$
> > > > >
> > > > > $q(z=2|s) = 0.2x + 0.9y + 0.9z$
> > > > >
> > > > > where $x = p(y=1|s), y = p(y=2|s),z = p(y=3|s)$,
> > > > >
> > > > > you end up at $x=5$, and $y + z = 5$, so any combination of $y + z$ (even if they are positive integers only) satisfies the equality.
> > > > >
> > > > > This example serves to illustrate the more general case where there are more number of classes than number of experts.

---

> > > > > > ### Comment · AnonReviewer1 · 2020-11-24
> > > > > > **Is this general?**
> > > > > >
> > > > > > Is this not a special case due to $q(z=1|y=2) = q(z=1|y=3)$? If you changed is so those values were all unique (which the would be in general I think) would that not provide unique (integer) solutions. For example a slight change to $q(z=1|y=1) = 0.7, q(z=1|y=2)=0.2$ would make the integer solutions unique again.
> > > > > >
> > > > > > And even in this example we have already exposed $p(y=1|S)$ correctly and uniquely to the server.

---

> > > > > > > ### Author Response · Authors · 2020-11-24
> > > > > > > **No, it is not**
> > > > > > >
> > > > > > > Thank you for pointing this out - indeed, we found a few cases where with the additional constraint that all per-data-points need to add up to the total number of data-points as well as that each individual p(y=c|s) needs to be positive, some systems of equations are solvable, whereas others are not.
> > > > > > > In the example above with $q(z=1|y=1) = 0.8, q(z=1|y=2) =0.6, q(z=1|y=3)=0.4$, there are multiple solutions $(x, y, z) \in [(3, 7, 0), (4, 5, 1), (5, 3, 2), (6, 1, 3)]$.
> > > > > > > It goes beyond this work (and this rebuttal) to fully characterise under which conditions the marginal can be reconstructed and which not. We updated the corresponding section in our paper.
> > > > > > >
> > > > > > > We thank you for sticking with us - our paper became much more precise about its privacy implications.

---

### Official Review · AnonReviewer2 · 2020-11-05
**Interesting approach, but subpar results and needs more clarification**

**Rating:** 4
**Confidence:** 4

**Review:**

-- Summary --
The paper proposes a method for federated learning of a mixture of experts model (FedMix). The approach allows training an ensemble of models each of which specializes to a subset of clients with similar data characteristics. The authors argue that this way of training an ensemble reduces the gradient divergence/interference, improves the overall performance, and sometimes reduces the communication overhead. The new method is evaluated a few federated image classification datasets.


-- Overall evaluation --
I find the idea of training a model ensemble instead of a single global model extremely appealing in the context of federated learning. The fact that the proposed approach allows models in the mixture (i.e., experts) to automatically specialize to different subsets of clients throughout training without having to run any clustering on the server makes the approach particularly appealing. Having said that, I also believe that the paper in the current form has many weaknesses, including:
- a significant lack of clarity throughout section 2 (see comments and questions below),
- in terms of methods, fairly ad-hoc changes introduced into the evidence lower-bound objective given in eq. 5 (removal of the entropy term and addition of a different regularize, which is justified in a very handwavy way),
- weak experimental results, which indicate that FedMix is, in fact, worse than the baselines both in terms of performance and the communication cost (unless I'm misreading Table 1 and Figure 4).

I believe that the paper has interesting and novel ideas, but falls short on both presenting them as well as getting them to work empirically. I would not recommend accepting the paper in the current form.


-- Comments and questions --

- Using MoE in federated learning makes a lot of sense intuitively. The authors further specialize MoE to FL by introducing a local gating network and then tying everything together through a global q distribution that approximates the posterior. These design choices are justified by stating that running federated optimization on the objective given in eq. 3 does not work well. I wonder, however, why not use a shared gating network instead of having to approximate the objective?

- I find the way lower bound in eqs. 5-6 is introduced and then the entropy term is dropped and substituted with some regularize (see 2nd paragraph on page 4) very ad-hoc. Typically, one would use a graphical model such as the one given in Figure 2 to derive a loss function in a principled way, without additional "hacks." Why use the graphical model formalism in the first place, if the final loss is being designed using some additional intuitive heuristics? Relatedly, perhaps there's a way to change the graphical model somehow and show that the final loss corresponds to the evidence lower bound in a slightly different probabilistic model? (e.g., with a different prior)

- Since $q_\phi(z \mid y)$ is a distribution over the experts in the mixture conditional on the class of the data point, what happens if K > # classes? It seems like q will select a separate model for each output class, so each expert will specialize to a single class only. How would that affect performance? Is K < # classes a requirement?


-- Experiments --

- If I'm not misreading the results in Table 1, FedMix adds a drastic communication overhead (due to multiple experts in the mixture) and does not perform as well as biased FedAvg on the CIFAR 10/100 datasets. Results for two baselines for EMNIST are missing.

- Similarly, results presented on Figure 4 for CIFAR datasets demonstrate that the baselines are better.

- For FedAvg, was the accuracy computed for the global model or for the global model after fine-tuning locally? Since FedMix targets to improve personalized FL, it should be compared with FedAvg with a simple fine-tuning-based personalization.


-- Minor points --

- Figure 1: are the bold curves some kind of running averages? Also, if the figure is given in the main text, it's strange that the definition of gradient divergence or alignment is provided in the appendix. I would recommend giving it somewhere in the main text.
- Figure 3: which dataset and task are these results for?
- Section 4, paragraph 1: I believe the EMNIST dataset has 3400 users if the version from tensorflow federated is used.
- Very minor: In Eq. 3, the notation for the sum over S is strange; maybe change to $\sum_{s=1}^S$?

---

> ### Author Response · Authors · 2020-11-19
> **Response to AnonReviewer2**
>
> We thank the reviewer for their thorough review. We are glad to see that you find our approach and its specialization capabilities extremely appealing. We would like to address your concerns.
>
> Indeed, optimizing the MoE likelihood directly is challenging, which we first observed in the centralized training case. Before deciding upon localized or shared gating networks, performing gradient ascent on the likelihood directly is challenging to get to work in practice. Our motivation of lower-bounding the likelihood is therefore related to the MoE model, and not necessarily to the federated scenario. We find that the gate immediately collapses to a single expert and does not recover. Lower-bounding proved to be a robust solution to this problem in addition to the benefits we gain in the federated setting. We have added a visualization of the MoE behavior to the Appendix H.
>
> You mention our changes to the lower-bound in the form of removing the entropy term at the client level. Indeed, our motivation for this is that we would like to encourage specialization of experts. As FedAvg shows, a single expert is quite capable of modelling the non-iid dataset to some degree. In order to not train ‘K almost identical experts’ and to manifest our intuition about the role of experts, we target specialization by encouraging low-entropy for the gates. Empirically we show that this intuition leads to better models (Figure 3). We believe that it is better to make such a decision ‘informed’ by first discussing the graphical model in its entirety. Similarly, the additional entropy term at the server is used since without an entropy term at the client, we empirically observed that some experts were not used at any shard. We understand that these modifications stem from intuition and empirical observations, which is why we either experimentally show the consequences of each modification (e.g. Figure 3) or mention what we observed directly.
>
> K<#classes is not a requirement, Figure 7 a) and c) in the Appendix show K=10 for Cifar10. We have updated the plots to include experiments with K=12 and K=14 for completeness.
>
> With respect to the discussion of our experiments. It is true that FedMix adds a communication overhead to FedAvg as well as being out-performed by the specialized biased FedAvg approach that targets label skew. We argue that FedMix offers a general method, and, by marginalizing across non-iid dataset configurations, so to say, offers a robust choice.
> All learning curves are created by evaluating the local dataset on the most recent server model (i.e. without fine-tuning). We observe significant overfitting in all considered algorithms when fine-tuning locally. Based on your suggestion, we have added a discussion on this point in the Appendix.
>
> With respect to your minor points:
> Yes, our Figures display the average of a sliding window. We have updated the main text accordingly.
> Figure 3 contains result on cifar10, we updated our caption.
> We are using the ‘Femnist’ dataset as provided here https://github.com/TalwalkarLab/leaf, consisting of 3.5k users.
> We have fixed notation
>
> Thank you for pointing out ways to improve our paper. We hope that with the additional clarifications and modifications you will consider raising your score.

---

### Author Response · Authors · 2020-11-19
**To all reviewers**

We thank the reviewers for their thorough evaluation and their feedback. We have updated the submission and made the following changes:
1. In addition to experiments on non-i.i.d-ness on $p(y)$ and $p(x)$, we present a discussion on the experimental setup of Clustered Federated Learning [Sattler et. al,2019] and show that FedMix is able to perfectly recover the label permutation cluster assignment across the different clients. This corresponds to non-i.i.d-ness in $p(y|x)$.
2. We have changed our parameterization of the gating mechanism $p(z|x,s)$ for the Femnist experiments to be consistent across all experiments. We have updated tables, plots and the discussion of those results.
3. We added, in the Appendix, a discussion on the potential danger of inferring the label marginal $p(y|s)$ in FedMix and FedAvg.
4. We discuss overfitting when locally fine-tuning models in the Appendix.

We believe your suggestions and critique allowed us to improve our submission and hope that you consider increasing your score.

---

### Author Response · Authors · 2020-11-24
**Towards the end of the discussion period**

Given that there are approximately 24h left until the end of the discussion period, we hope that the reviewers had the chance to read our rebuttal and revise their opinions. We are thankful for your considerations and hope that we could address your concerns. If so, we would appreciate a comment and reevaluation of your score to better reflect the current state of our submission.

---

### Decision · Program_Chairs · 2021-01-07
**Final Decision**

**Decision:**

Reject

**Comment:**

This paper presents an approach, FedMix, for federated learning using mixture of experts (MoE). The basic idea is to learn an ensemble of models and user-specific combination weights (mixing proportions).

The reviewers appreciated the MoE formulation for federated learning. However, there were multiple concerns from the reviewers, which include lack of clarity (regarding the variational lower bound that is being used), significant communication cost and privacy concerns (the server can infer the users' label distributions), weak experimental results, and lack of any theoretical support (which isn't that big an issue if the paper were stronger on other aspects). The author feedback was considered and the reviewers engaged in some discussions (also with the authors). In the end, however, the reviewer were still not convinced that the paper is ready to be published in its current state. Based on my own reading of the paper, the reviews, and the author response, I agree with this assessment.

The authors are advised to take into account the feedback from the reviewers and resubmit to another venue.